# Fatigue Crack Growth in Welded S355 Samples Subjected to Bending Loading

**Janusz Lewandowski [1],* and Dariusz Rozumek [2]**

1. Measurement and Automation Center S.A., 41-800 Zabrze, Poland
2. Opole University of Technology, 45-758 Opole, Poland; d.rozumek@po.edu.pl
* Correspondence: janusz210@wp.pl; Tel.: +48-604-58-52-04

**Abstract:** The paper presents a result of experimental tests of welded S355 samples subjected to bending loading. In order to analyze how the fillet joints' shape and the load ratio affect the crack growth, we selected two kinds of the fillet shape: concave and convex, and load ratios, namely R = −1. Samples with stress concentrators in form of the two-sided fillet welded joint were tested. The test results were compared to experiments conducted on welds samples with and without heat treatment.

**Keywords:** fatigue crack growth welding; fillet welds; bending; micro hardness; residual stress; microstructure

## 1. Introduction

The development of civilization is extremely intense nowadays. Compared to the previous century, mankind has made tremendous progress in many technical disciplines. Despite the most dynamic developing areas of the so-called new technologies, the classic problems of fracture mechanics and fatigue strength are still relevant today. It can be said that they constitute a stable foundation for further and safe development. The durability of welded joints in engineering structures is the subject of many studies and scientific publications [1–5]. Knowledge in this field, taking into account the type of welded joint, the microstructure of individual zones of the joint, the impact of heat treatment, the impact of changes in hardness or the type and distribution of stresses, is crucial so that the resulting structures are safe and durable. Welded methods and technologies are constantly being improved. Unfortunately, joining by welding, despite its advantages, also has disadvantages. Regardless of the welding technology, heat is always introduced into the joined elements and they are heated to high temperatures, which contributes to changes in the structure of the welded joint area and the heat-affected zone (HAZ), resulting in the formation of the previously described structural notch. If the welded joint is designed in such a way that the face and/or the ridge of the weld cause a change in the geometry of the element, a geometric notch is also created. Both resulting notches negatively affect the fatigue life, and stress concentrations usually occur in these areas. Another problem affecting the fatigue life of welded joints is the thermal processes occurring in the joint during welding. It can be said that in the welded joint and in the heat-affected zone, metallurgical processes take place, resulting in structures that differ significantly from the parent material (martensite, bainite, sorbite, recrystallization zones) [5–7].

The aim of this work is to present the results of the fatigue crack development of T-shaped welded joints with fillet welds made of steel S355 subjected to bending loads, taking into account the shape of the welds and heat treatment.

## 2. Materials and Methods

The test specimens were made of structural steel of increased strength, grade S355, in the normalized state. This steel is widely used in industry, including the construction of ships, bridges, lifting devices, tanks and pipelines, etc. Chemical composition of the used

material shows in Table 1 and the most important mechanical properties in Table 2. The starting material of the specimens was a drawn rod with a diameter of Ø 30 mm. Then, as a result of the performed machining and the TIG welding process (TIG welding is the production of an electric arc using a non-consumable tungsten electrode in an inert gas shield), ready-made samples were obtained. T-welded joints, with fillet welds, were made in two variants of the weld face, i.e., concave and convex.

**Table 1.** Chemical composition of the S355 steel (in wt %).

| C | Mn | Si | P | S | Cr | Ni | Cu | Fe |
|---|---|---|---|---|---|---|---|---|
| 0.2 | 1.49 | 0.33 | 0.023 | 0.024 | 0.01 | 0.01 | 0.035 | Balance |

**Table 2.** Mechanical properties of the S355 steel.

| Yield Stress, $\sigma_y$ (MPa) | Ultimate Stress, $\sigma_u$ (MPa) | Young's Modulus, E (GPa) | Poisson's Ratio $\nu$ (−) | Elongation, $A_5$ (%) |
|---|---|---|---|---|
| 357 | 535 | 210 | 0.30 | 21 |

The experimental tests were carried out on samples welded without heat treatment and on samples after two different heat treatments. The first heat treatment was performed by subjecting the samples to annealing at the temperature of 630 °C for 2 h. However, the second heat treatment consisted of normalizing annealing at the temperature of 940 °C for 1.1 h [8–15]. Geometries of the tested samples are presented in Figure 1.

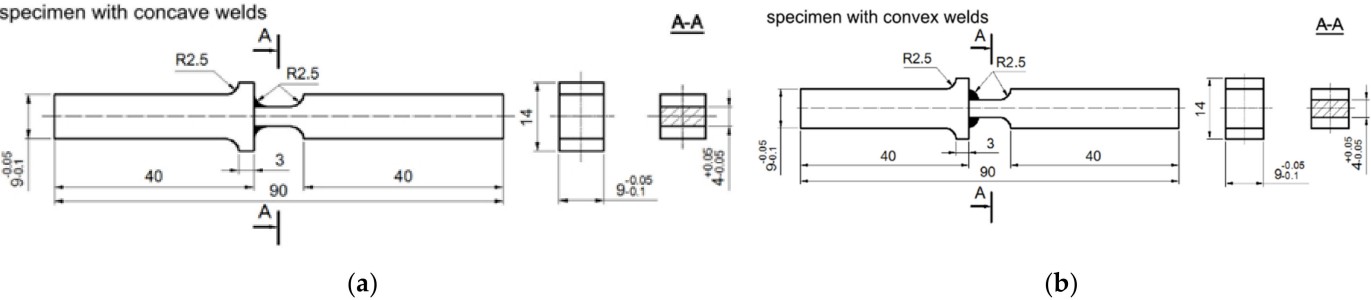

(a)          (b)

**Figure 1.** Geometries of specimen: (**a**) with concave welds, (**b**) with convex welds, (dimensions in mm).

Then, in accordance with the requirements of EN ISO 9015-1, hardness measurements on the Vickers scale were carried out using a LECO MHT 200 hardness tester (LECO Corporation, St. Joseph, MO, USA), under a load of 100 g.

Metallographic tests were performed with the use of an optical microscope OLYMPUS IX70 (Olympus Corporation, Japan). The test to fatigue crack growth under cyclic bending were performed in the laboratory of the Department of Mechanics and Machine Design at Opole University of Technology on the fatigue test stand MZGS-100 [15,16] (Figures 2 and 3). The tests were conducted under amplitude of total force moment control with the loading frequency 28.4 Hz. The scope included fatigue tests under low-cycle fatigue (LCF) and high-cycle fatigue (HCF) conditions. The samples restrained on one side were loaded with a constant amplitude of moments with the value Ma = 9.2 N·m and the load ratio R = −1. The theoretical stress concentration factor, estimated with use of the model [17], in the solid specimen with concave weld under bending was $K_t$ = 1.38 while it was 1.56 for the convex weld configuration.

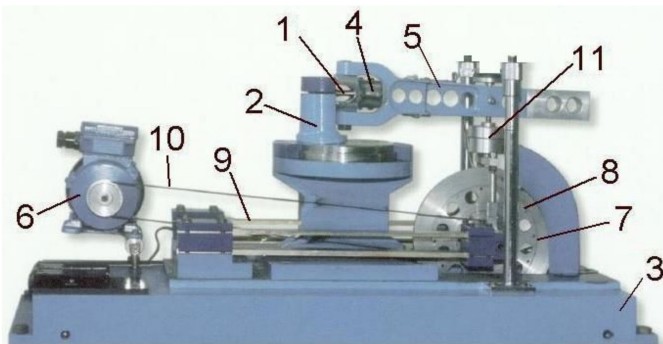

**Figure 2.** MZGS-100 machine, where: 1—specimen, 2—rotational head with a holder, 3—bed, 4—holder, 5—lever, 6—motor, 7—rotating disk, 8—unbalanced mass, 9—flat springs, 10—driving belt, 11—hydraulic connector.

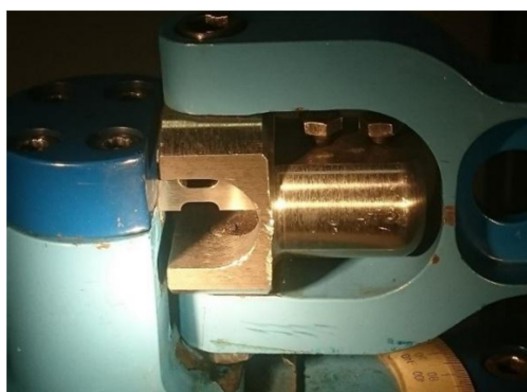

**Figure 3.** Clamped specimen in MZGS-100 machine.

The fatigue crack growth was measured with a micrometer placed in a portable optical microscope with a 20-fold magnification and an accuracy of 0.01 mm. The number of load cycles N was also recorded for each measurement point.

### 3. Results

Selected solid and welded samples were subjected to hardness tests using the Vicersa HV0.1 method, in accordance with the EN ISO 9015-1 standard. The results of hardness measurements of welded specimens without heat treatment (HT) and after HT are shown in Figure 4 for the averaged results of samples with concave and convex welds. For the native material of the specimens, the hardness values remained at the level of (188–189 $HV_{0.1}$). However, for specimens welded without HT, the hardness values changed significantly depending on the place of measurement. For the base material, the hardness remained the same (188–189 $HV_{0.1}$). While in the heat-affected zone (HAZ) large fluctuations in hardness were observed (194–248 $HV_{0.1}$). Then, moving to the work piece, the measured values decreased and stabilized (230–220 $HV_{0.1}$). In the specimens subjected to relief annealing (temperature 630 °C and time 2 h), the measured hardness and their variability were lower compared to the hardness of the specimens without HT. However, in the specimens subjected to normalizing annealing (temperature 940 °C and time 1.1 h), the hardness and their fluctuations reached the lowest values, compared to all tested specimens.

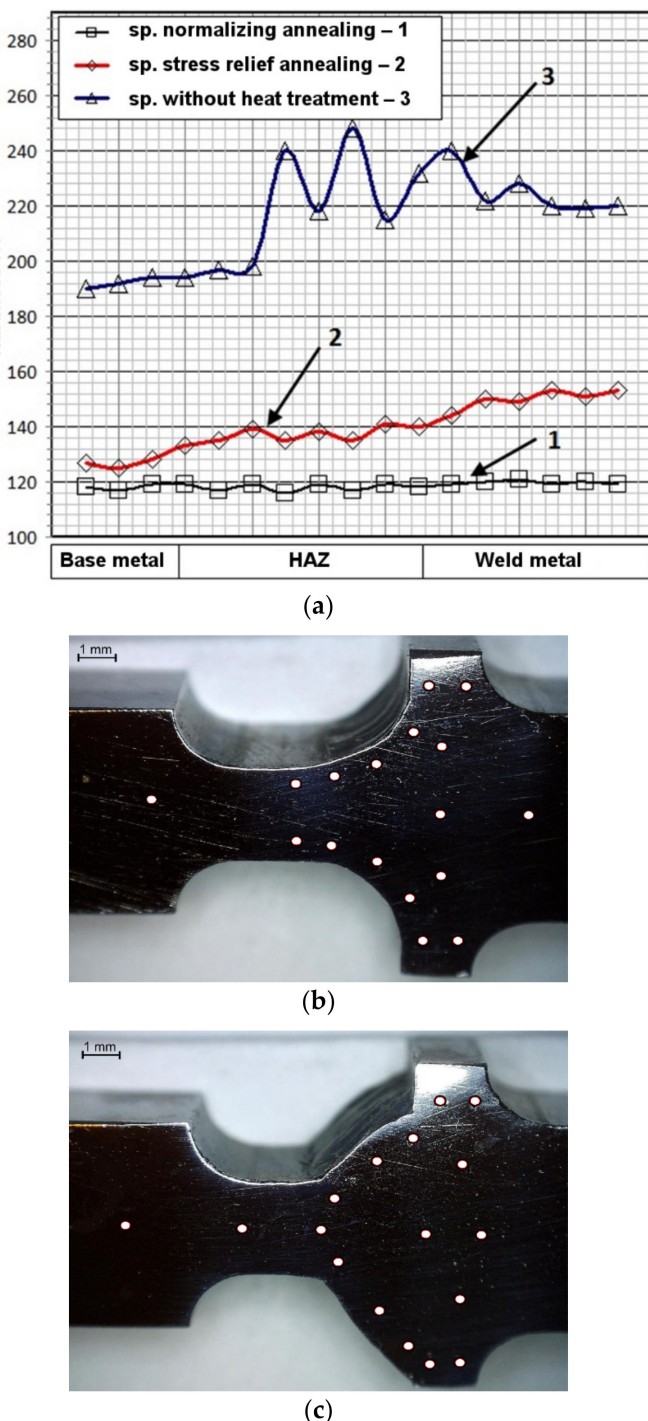

**Figure 4.** Hardness values including the location of the measurement in the samples, (**a**) hardness diagram; (**b**) place of measurement in strokes with concave welds; (**c**) place of measurement in strokes with convex welds. HAZ: heat-affected zone.

The metallographic tests of the base material confirmed the ferritic–pearlitic microstructure with a band pattern (base—Figure 5). The base material was characterized by ferrite grains with a diameter of 5 to 40 μm and similar grain sizes of perlite with a diameter of 5 to 45 μm.

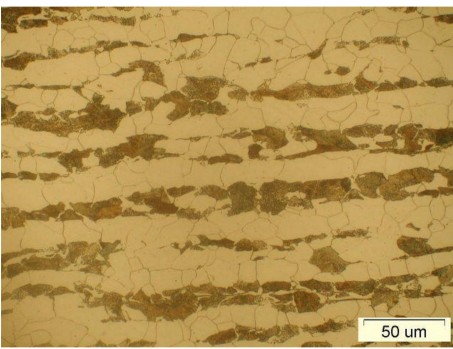

**Figure 5.** Steel structure base metal.

Figure 6 shows the zoning image observed in the test specimens without HT (Base metal, HAZ, Weld metal). The base material has the typical ferritic–pearlitic microstructure of the material used. In the heat-affected zone, microstructures can be distinguished: partially recrystallized, fine grained normalized zone, lower bainite zone, upper bainite and martensite structure. On the other hand, the microstructure of the weld was dendritic with grains in the Widmanstatten system.

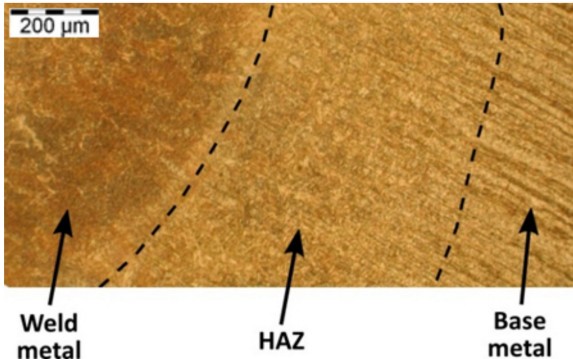

**Figure 6.** Three regions of samples welded without heat treatment: the base material, heat-affected zone (HAZ) and the weld region.

The microstructure of the material of the welded specimen subjected to relief annealing is shown in Figure 7. In the heat-affected zone there is a coarse-grained structure of bainite and sorbite.

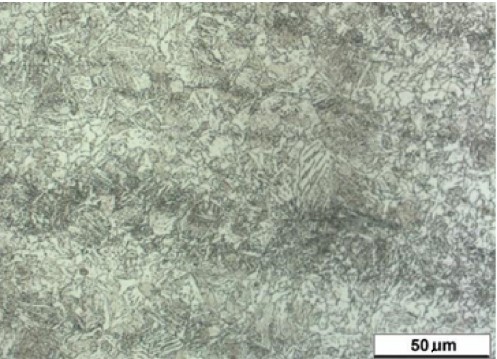

**Figure 7.** Microstructure of the welded specimen after relief annealing (500× magnification), bainite and sorbite structure.

The microstructure of the material of the welded specimen subjected to normalizing annealing is shown in Figure 8. In the case of normalizing annealing in the weld and HAZ, a coarse-grained ferrite structure with a small amount of pearlite was formed.

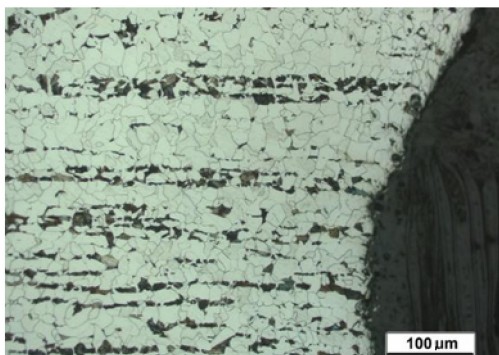

**Figure 8.** Microstructure of the welded specimen after normalizing annealing, ferrite with a small amount of pearlite.

Additional stresses appear in the material of welded specimens as a result of uneven solidification of welds and the occurring structural changes. Residual stresses (RS) do not depend on the action of external forces. Tensile stresses occur in the weld face zone, and compressive stresses in the HAZ zone. Measurements were made using the strain gauge and trepan method in accordance with the standard (ASTM E837-08). The stress distribution in the tested specimens is shown in Figure 9. The tensile stress values reached their maximum values up to 280 MPa in the center of the welds for joints without HT. On the other hand, the compressive stresses in the HAZ regions. These stresses can cause deformation of the workpiece. The distribution of stresses and their values in samples with convex and concave welds was similar.

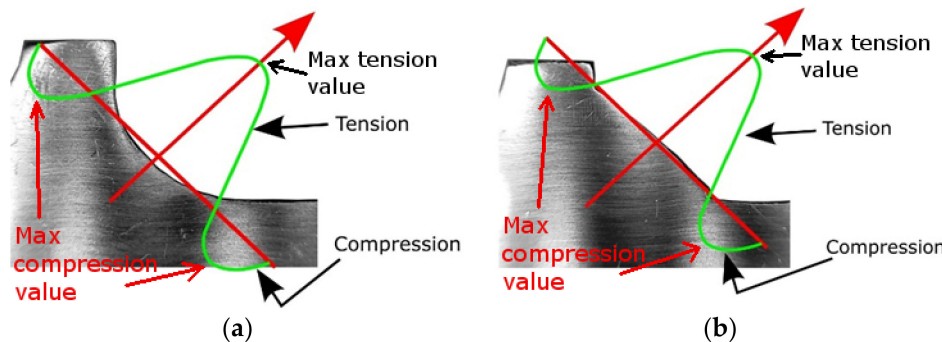

**Figure 9.** The distribution of residual stresses in the tested T-joints with fillet welds, where: (**a**) samples with concave welds, (**b**) samples with convex welds.

Figure 10 shows the fatigue crack lengths "a" as a function of the number of cycles "N" for the tested samples, the values of which were closest to the average of the five tested samples. The scatter of the results ranged from 2 to 3%. From Figure 10a, the lowest fatigue life for the specimen after normalizing and concave welds can be seen. Crack initiation (0.10 mm) was at 4700 cycles. The further development of the crack was rather quick and the specimens failed at 7000 cycles. In the specimens subjected to relief annealing, the initiation of fatigue cracks occurred at the number of cycles of 15,100 cycles, and the failure of the specimens occurred at the number of cycles of 19,000. The specimens without heat treatment showed the highest fatigue life, in which the crack initiation took place at the number of cycles of 69,000, and their failure at the number of cycles of 77,500.

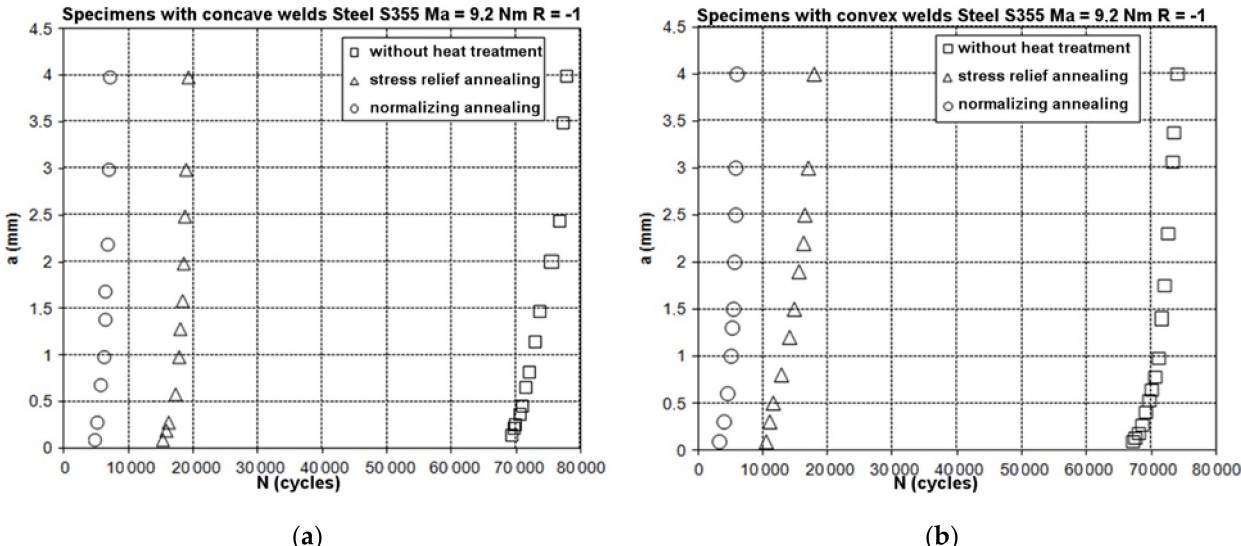

**Figure 10.** Fatigue crack length vs. number of cycles under bending for specimens with: (**a**) concave welds, (**b**) convex welds.

Figure 10b shows the results of fatigue tests for specimens with convex welds, where it can also be seen that in this case the lowest fatigue life was shown by the specimens after normalizing annealing. Crack initiation (0.10 mm) was at 3200 cycles. Further growth of cracks occurred at a rapid pace and the specimens failed at 5900 cycles. In the samples subjected to relief annealing, the initiation of fatigue cracks took place at the number of cycles of 10,500 cycles, and the failure of the specimens occurred at the number of cycles of 18,000. Samples without heat treatment, as in the case of specimens with concave welds, showed the highest fatigue life, where the crack initiation took place at the number of cycles of 67,000, and their failure at the number of cycles of 74,000.

The differences in the fatigue life of the tested specimens with concave and convex welds are significant. In the case of specimens with concave welds, the decrease in fatigue life of relief annealed specimens was 75%, and for specimens annealed normalizing it was 90% compared to specimens without heat treatment. As in the case of specimens with concave welds, a 76% decrease in fatigue life of relief annealed specimens was observed for samples with convex welds, and 92% for specimens with normalizing annealing compared to specimens without heat treatment. When comparing the durability of samples with concave and convex weld faces, for the same specimen states (without heat treatment and after heat treatment), it can be seen that for specimens with concave welds, the durability was always higher compared to specimens with convex welds. The decrease in fatigue life of specimens with convex welds without heat treatment and relief annealing was 6%, and for specimens with normalizing annealing, it was 15% in comparison to specimens with concave welds. According to the authors, the significant drops in fatigue life in the specimens subjected to heat treatment were caused by structural changes taking place in the tested material. On the other hand, the higher durability of specimens with concave welds compared to specimens with convex welds was due to the occurrence of sharp notches in specimens with convex welds, which gave rise to cracks.

Exemplary fatigue crack path is shown in Figure 11 for a specimen with concave welds (without HT) and in Figure 12 for a specimen with convex welds under bending (without HT). During experimental tests, the crack initiation occurred on one side of the sample, most often on the upper surface and less often on the lower surface. A single crack propagated to a certain value, after which a crack also appeared on the other side of the sample.

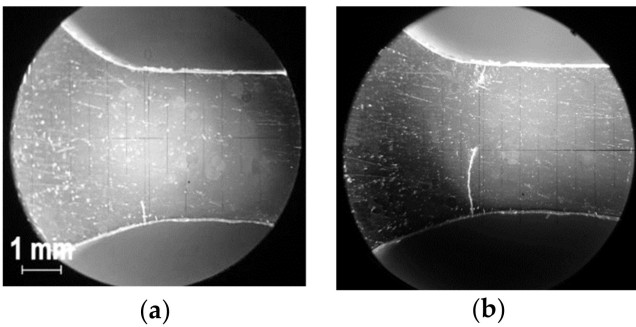

**Figure 11.** Selected stages of crack development in a specimen with concave welds in bending without HT (heat treatment), where: (**a**) crack initiation; (**b**) crack initiation and propagation.

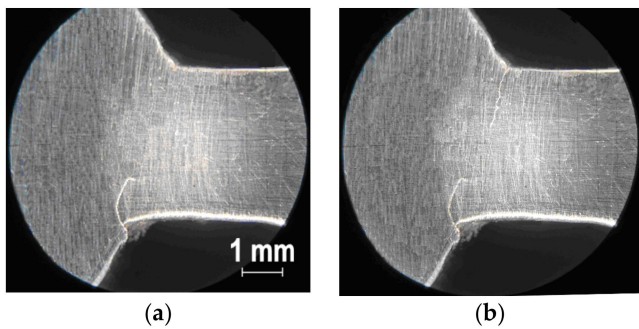

**Figure 12.** Selected stages of crack development in a specimen with convex welds in bending without HT, where: (**a**) crack initiation; (**b**) crack initiation and propagation.

Figure 13 shows a damaged specimen after relief annealing. The main fracture is located outside the weld and the original HAZ in the area of the sample with a spheroidite structure, strip enriched with fine-grained cementite (carbides). From the main fracture, numerous lateral fractures initiated in areas rich in carbide precipitates developed.

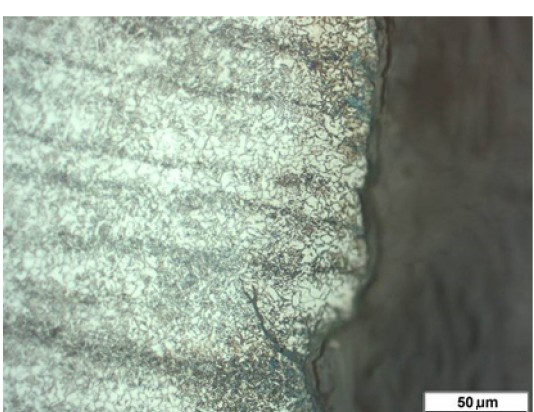

**Figure 13.** Development of secondary cracks.

A sample welded after normalizing annealing is shown in Figure 14. The main crack propagation took place outside the weld in a homogeneous structure of the ferritic–pearlitic zone of the HAZ. There are also side cracks propagating along the grain boundaries. Within the scrap, mixed cracking is observed: brittle and plastic with areas of high plastic deformation.

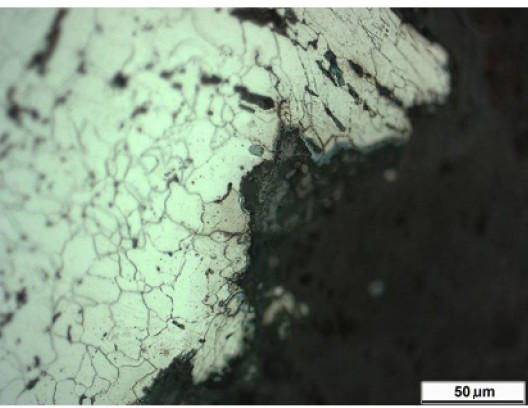

**Figure 14.** Damaged welded sample after normalizing.

## 4. Conclusions

On the basis on the obtained results of experimental research and observations, the following conclusions can be formulated:

- The best life was demonstrated by specimens without heat treatment with slightly higher durability of specimens with concave welds;
- Fatigue crack growth paths in all test samples started on one side, most often on the upper surface and less often on the lower surface. As the cracks were growing, cracks also appeared on the other side of the specimen;
- Tested samples, with and without heat treatment, show different fatigue fractures;
- The initiation and propagation of cracks usually occurred in the HAZ where the highest micro-hardness was measured;
- The highest material hardness was measured on specimens without heat treatment in HAZ, and the lowest in specimens after normalizing annealing;
- Specimens after relief annealing are characterized by greater brittleness, while after normalizing annealing more plasticity.

**Author Contributions:** Conceptualization, J.L.; formal analysis, J.L. and D.R.; investigation, J.L.; methodology, J.L.; writing—original draft, J.L. and D.R. All authors have read and agreed to the published version of the manuscript.

**Funding:** This research received no external funding.

**Institutional Review Board Statement:** Not applicable.

**Informed Consent Statement:** Not applicable.

**Data Availability Statement:** Not applicable.

**Conflicts of Interest:** The authors declare no conflict of interest.

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
