# Peer review of "Fatigue Crack Growth in Welded S355 Samples Subjected to Bending Loading"

_metals, doi:10.3390/met11091394_

Round 1

Reviewer 1 Report

The manuscript requires some changes before publication, see notes below. The disadvantage is (in the current form of the presentation) that the assessment of fatigue life is based only on the results shown in Fig.9. They were obtained on only one specimen for the three states (without heat treatment, with stress relief annealing and normalizing annealing) and for concave welds and convex welds. Since fatigue life always has a certain scatter, it would be highly desirable to report the results obtained on at least 3 test specimens for each state.

Notes:

line 55: „were obtained“ should be removed

line 74: The sentence should be explained in more detail

line 94: The sentence is incomprehensible

Fig. 5: lacks a scale

line 105: “sorbitol” means sugar alcohol with a sweet taste

line 117: “… more frequently…”  does it mean that it is not the case in some cases?

Line 147: Is the difference in fatigue life (7000/5900, 19000/18000, 77500/7400) really significant?

line 192: Why did the cracks always initiate on one side? Doesn't this indicate load asymmetry?

line 195: It is stated here that „… show different fatigue fractures“. However, different types of fractures are not mentioned in the article at all

Author Response

Dear Editor,

Enclosed you will find responses to the Reviewer comments of the manuscript entitled: Fatigue crack growth in welded S355 specimens subjected to bending loading for Journal Metals.

Responses to the Reviewer comments:

  1. .. The disadvantage is (in the current form of the presentation) that the assessment of fatigue life is based only on the results shown in Fig.9.

Changed the sentence from:

Figure 9 shows the fatigue crack lengths "a" as a function of the number of cycles "N" for the tested samples.

on:

Figure 9 shows the fatigue crack lengths "a" as a function of the number of cycles "N" for the tested samples, the values of which were closest to the average of the five tested samples. The scatter of the results ranged from 2 to 3%.

  1. Changed the sentence from:

The test to fatigue crack growth under cyclic bending, were obtained were performed in the laboratory of the Department of Mechanics and Machine Design at Opole University of Technology on the fatigue test stand MZGS – 100 [13,14] (Figure 2).

on:

The test to fatigue crack growth under cyclic bending, were performed in the laboratory of the Department of Mechanics and Machine Design at Opole University of Technology on the fatigue test stand MZGS – 100 [13,14] (Figure 2).

  1. Changed the sentence from:

The results of hardness measurements of welded specimens without heat treatment (HT) and after HT are shown in Figure 3 for the averaged results of samples with concave and convex spins.

on:

The results of hardness measurements of welded specimens without heat treatment (HT) and after HT are shown in  Figure 3  for  the  averaged  results  of  samples  with  concave  and  convex  welds.

  1. Changed the sentence from:

Base material for the microstructure of the material used. In the heat-affected zone, microstructures can be distinguished: partially recrystallized, fine grained normalized zone, lower bainite zone, upper bainite and martensite structure.

on:

The base material has the typical ferritic-pearlitic microstructure of the material used. In the heat-affected  zone,  microstructures  can  be  distinguished:  partially  recrystallized,  fine grained normalized zone, lower bainite zone, upper bainite and martensite structure.

  1. Figure 6 has been changed and a scale has been added

  1. Changed the sentence from:

The microstructure of the material of the welded specimen subjected to relief annealing is shown in Figure 6. In the  heat affected zone there is a coarse-grained structure of bainite and sorbitol.

on:

The microstructure of the material of the welded specimen subjected to relief annealing is shown in Figure 6. In the heat affected zone there is a coarse-grained structure of bainite and sorbite.

  1. Changed the sentence from:

Tensile stresses occur most frequently in the weld face zone, and compressive stresses in the HAZ zone.

on:

Tensile stresses occur in the weld face zone, and compressive stresses in the HAZ zone.

  1. Line 147: Is the difference in fatigue life (7000/5900, 19000/18000, 77500/7400) really significant?

The article presents the results of experimental tests which showed differences in fatigue life for various heat treatments. The differences are greater for some cases and smaller for others. The results obtained on 5 - 6 samples for each case confirm the differences. Obviously, if a treatment with other parameters is performed, the results may vary.

  1. line 192: Why did the cracks always initiate on one side? Doesn't this indicate load asymmetry?

Crack initiation always occurred on one side, randomly: from the top or bottom of the sample. This could be due to the differences in the manual execution of welds, their geometry, penetration depth, etc.

  1. line 195: It is stated here that „… show different fatigue fractures“. However, different types of fractures are not mentioned in the article at all

The tested samples, with and without heat treatment, show various fatigue cracks. The authors meant the differences in material fracture without HT and with HT, brittle and plastic fracture.

The authors would like to thank the Reviewer for all their comments and detailed language remarks, which contributed to the increased value of our paper.

Your faithfully,

Janusz Lewandowski

Opole University of Technology

Reviewer 2 Report

Please check the attached file.  The comments were given there.

Author Response

Dear Editor,

Enclosed you will find responses to the Reviewer comments of the manuscript entitled: Fatigue crack growth in welded S355 specimens subjected to bending loading for Journal Metals.

Responses to the Reviewer comments:
I tried to make corrections in the content of the article, in accordance with the guidelines.

Unfortunately, I am not able to put other Figures 13 and 14. The available equipment allowed me to take photos in the quality presented in the article.

The authors would like to thank the Reviewer for all their comments and detailed language remarks, which contributed to the increased value of our paper.

Your faithfully,
Janusz Lewandowski
Opole University of Technology

Round 2

Reviewer 2 Report

The paper has been revised well.  In my opinion the paper is worth publishing.